# Plastic Deformation Mechanism of High Strength and Toughness ZK61 Magnesium Alloy Plate by Multipass Horizontal Continuous Rolling

**DOI:** 10.3390/ma16031320

**Published:** 2023-02-03

**Authors:** Ming Chen, Cong Ma, Qingjie Liu, Ming Cheng, Haolei Wang, Xiaodong Hu

**Affiliations:** 1School of Mechanical Engineering and Automation, University of Science and Technology Liaoning, Anshan 114051, China; 2Research Center of Magnesium Alloy Casting and Rolling Technology, University of Science and Technology Liaoning, Anshan 114051, China; 3Shi-changxu Innovation Center for Advanced Materials, Institute of Metal Research, Chinese Academy of Sciences, Shenyang 110016, China

**Keywords:** symmetric hot rolling, asymmetric warm rolling, finite element, viscoplastic self-consistent model, texture evolution

## Abstract

ZK61 magnesium-alloy plate with high tensile strength and elongation is obtained by combined multipass symmetric hot rolling and asymmetric warm rolling. Deformation history considering varying strain rate obtained from the macro-finite element analysis of the selected passes are introduced into the viscoplastic self-consistent model (VPSC) as initial boundary conditions for macro- multiscale and micro-multiscale coupling analysis. VPSC simulation results show that in the initial stage of rolling deformation, the basal <a> slip is the dominated deformation mode, supplemented by prismatic <a> slip and pyramidal <c+a> slip. With increased rolling strain, the pyramidal <c+a> slip presents competitive relationship with basal <a> slip, and the activation amount of {10$\overset{—}{1}$1} compression twins is limited. During asymmetric rolling, the basal <a> slip is dominant, followed by the pyramidal <c+a> slip. Experimental results show that the basal texture is gradually strengthened after symmetric rolling, and grain size is refined due to the activation and recrystallization of twins. Asymmetric rolling makes the basal texture deflect 10° to the rolling direction and further refine the grain size. With the ongoing of symmetric rolling, the mechanical anisotropy of the plate weakens, and the yield strength, tensile strength, and plasticity of the material improves. In particular, after asymmetric rolling, the tensile strength in the RD and TD directions of the plate reaches 391.2 MPa and 398.9 MPa, whereas the elongation reaches 19.8% and 25.5%.

## 1. Introduction

Advanced lightweight-material technology are an important core technology of “Made in China 2025”, and achieving carbon peaks is the latest goal of this technology. Studies have shown that for every 100 kg of weight loss, the fuel consumption per 100 km can be reduced by about 0.4 L, and the CO_2_ emissions can be reduced by about 1 kg [1,2]. Magnesium alloy is the lightest metal-engineering material, known as “the most promising green engineering material in the 21st century” [3,4]. Magnesium-alloy plate (thickness < 3 mm) has certain application prospects in the fields of new energy vehicles, military aerospace, and electronic 3C (Computer, Communication, ConsumerElectronics). As the initial blank of hollow thin-walled complex section parts, magnesium-alloy plate needs to have good formability at room temperature [5,6]. Thus, fabricating high-quality plates required by domestic magnesium-alloy plate rolling mills and meeting the demand for magnesium-alloy plates are urgent tasks for precision manufacturing. How to weaken the basal texture and refine the grains is the core issue in the research and development of high-quality magnesium-alloy plates. At present, AZ-based magnesium alloys have insufficient strength under high load, and rare-earth magnesium-alloy plates have sufficient strength [7,8,9]. Chen [10] studied the microstructure change of ME21 rare earth magnesium alloy based on VPSC model. An [11] studied the evolution law of dual-phase structure of ultra-thin Mg-Li alloy during asymmetric warm rolling. However, the latter are expensive and difficult to popularize. ZK-series magnesium alloys are the most promising high-strength magnesium alloys currently available in aerospace, military, new energy vehicles, and other fields. ZK61 magnesium alloys are gradually attracting attention from all sides because of their excellent strong plastic properties. Chen [12] studied the evolution of the microstructure and properties of magnesium-alloy plates during ZK61 magnesium-alloy rolling. Texture evolution depends on rolling parameters (reduction, temperature) and deformation [13,14]. Accordingly, the influence of texture of ZK61 magnesium alloy in rolling state on the microstructure and properties of plate rolling is particularly important. Simulating and predicting the influence of micro-deformation mode startup on texture and macro-mechanical properties are also significant to provide a theoretical basis for developing ZK61 magnesium-alloy plate with high strength and excellent mechanical properties in the future [15,16].

In the present study, as-cast untreated ZK61 magnesium-alloy plate with a thickness of 17 mm is used as the raw material for rolling experiments. A symmetric large-scale rolling mill and an asymmetric six-high warm rolling mill are used as the processing methods. Starting from rolling a slab with a thickness of 17 mm, seven passes of hot rolling to 3.4 mm are performed with a large rolling mill. Then, a 0.5 mm ZK61 magnesium-alloy plate is prepared by multiple passes of a six-high warm rolling mill with a small reduction. The finite-element method (FEM) is used to systematically study the multipass continuous hot rolling and asymmetric warm rolling of ZK61 magnesium-alloy plate, as well as to explore the variation law of the equivalent strain field during each rolling pass [17]. Accordingly, the plate plastic-strain data are extracted, i.e., the data from the SR-3.4 mm plate of symmetric hot rolling and the asymmetric rolling (SR), to obtain the 1.5, 0.75, and 0.5 mm-thickness plates. The rolled plates are sampled for electron backscatter diffraction (EBSD) testing and room-temperature tensile tests. Then, the viscoplastic self-consistent model (VPSC) model is used to simulate the rolling deformation [18,19,20,21] of the above-mentioned representative ZK61 plate. Through the combination of symmetrical rolling and asymmetric rolling experiments, the aim is to prepare magnesium alloy sheets with high mechanical properties, and the yield strength, tensile strength and elongation of the final sheets are greatly improved. The paper first introduces the theoretical content of the experiment and simulation, and then compares and analyzes the experimental and simulation results, and draws conclusions on this basis. Schematic diagram of symmetric rolling and asymmetric rolling and the stress state of the plate are shown in Figure 1. Due to the combination of symmetric hot rolling and asymmetric hot rolling, the production efficiency of this paper is not high.

## 2. Experimental

### 2.1. Experimental Materials and Methods

The materials used in the experiment are ZK61 as-cast plate with dimensions of 195 mm × 200 mm × 17 mm. Before the start of the rolling experiment, the plate needs to be homogenized and heat treated to eliminate the segregation of the intragranular structure [22]. The treatment temperature is 400 °C, and the time is 14 h. The equipment is LDK-A tunnel furnace, U-shaped tube is used for heating, and the maximum heating temperature is 500 °C. The chemical composition of ZK61 magnesium-alloy plate is shown in Table 1.

In the present study, the as-cast ZK61 magnesium alloy of a 17 mm-thick plate is first rolled by Symmetrical Rolling (SR) (Research Center of Magnesium Alloy Casting and Rolling Technology, University of Science and Technology Liaoning, Anshan, China), followed by large reduction, seven passes, and hot rolling to prepare 3.4 mm-thick magnesium-alloy rolled plates, total deformation is 80%. The rolled plates are heated to 380 ℃ before rolling, and after each pass of rolling, the finished product is immediately placed in a furnace for 15 min. Then, we use an asymmetric warm-rolling (ASR) experimental equipment (Research Center of Magnesium Alloy Casting and Rolling Technology, University of Science and Technology Liaoning, Anshan, China) comprising six high warm-rolling mills, The roll temperature is controlled at 280 °C, and the diameters of the work rolls selected in this ASR experiment were as follows: the upper roll diameter is 75 mm, and the lower roll diameter is 55 mm. Small reduction, multipass and reversible rolling are used to obtain a 0.5 mm plate, taking 3.4 mm plate as the initial plate, roll to 0.5 mm, and the total deformation reaches 85%. During rolling, representative samples of 1.5, 1.3, and 0.75 mm are collected. The roller speed is 1.197 rad/min. The specific procedures are shown in Table 2.

The XRD (University of Science and Technology Liaoning, Anshan, China) phase composition of the initial ZK61 magnesium alloy as-cast plates and the plates after homogenization heat treatment is analyzed and detected. Figure 2 shows the diffraction pattern of the plate detected by XRD experiment. Results reveal that after the homogenization heat treatment, the diffraction peak of MgZn_2_ in the billet is significantly reduced. This finding indicates that after the homogenization heat treatment, the ZK61 as-cast magnesium alloy is dominated by single-phase α-Mg, and the grain boundary precipitates basically disappear.

### 2.2. Establishment of VPSC Model and Determination of Related Parameters

The tiny grain distribution of the micro-orientation affects the formation of the texture evolution obtained by calculation through the macroscopic (FEM) and the microscopic VPSC model. The FEM grid type is quadrilateral grid, and the element type is set as thermally coupled CPE4RT element.The multiscale analysis of ZK61 magnesium-alloy plate rolling [23] is performed as follows.

The velocity gradient of the corresponding model is
(1)$$L={\left(\begin{array}{ccc} \dot{\varepsilon } & 0 & -\dot{\gamma }\\ 0 & 0 & 0\\ -\dot{\gamma } & 0 & -\dot{\varepsilon } \end{array}\right)}_{1-RD,2-TD,3-ND}$$

The above velocity-gradient expression shows that the rolled plate is stretched in the RD direction, compressed in the ND direction, and sheared in the TD direction during rolling is normal strain rate, and is the shear strain rate. The velocity gradient in the formula is expressed by the coordinate systems 1-RD, 2-TD, and 3-ND, i.e., the sample coordinate systems RD, TD, and ND are parallel to strain/stress main axes 1, 2, and 3, respectively.

According to the above assumptions and numerical requirements, the two-dimensional FEM is used to conduct macroscopic numerical simulation of the rolling process, thereby obtaining a relatively accurate strain-rate component. Only the SR-3.4 mm plate of SR and the representative 1.5 mm and ASR-0.75 mm plates selected for asymmetric rolling are simulated and analyzed. The texture changes and microstructure evolution of the plates after asymmetric rolling are analyzed relative to those of SR. Figure 3 and Figure 4 shows the stress field results and strain evolution of selected elements for SR-3.4 mm plate during simultaneous rolling. Considering the influence of variable strain rate in the process of rolling deformation, the variable strain rate is imported into the VPSC model as a boundary loading condition for simulation calculation.

Figure 5 shows the stress-field distribution obtained from the simulation of 1.5, 0.75, and 0.5 mm sample plates during the asymmetric rolling and the evolution of each strain component of the selected element inside the workpiece. PE11, PE12, and PE22 represent normal, shear, and reverse strains, respectively, and represents the strain rate corresponding with strain.

Considering the influence of the velocity-gradient tensor on the calculation accuracy of VPSC, the calculated strain-rate data of each pass is inputted into the VPSC model input file as the boundary loading condition.

### 2.3. Material Parameter Setting

At room temperature, ZK61 magnesium alloy primarily includes basal slip, prismatic slip, pyramidal slip, tensile twinning, and compression twinning to participate in the plastic deformation of the material [24]. At room temperature, the elastic constant formula of ZK61 magnesium alloy is as follows:(2)$$C_{ij}=\left(\begin{array}{cccccc} 56.3 & 22.33 & 22.33 & 0 & 0 & 0\\ 22.33 & 56.3 & 22.33 & 0 & 0 & 0\\ 22.33 & 22.33 & 56.3 & 0 & 0 & 0\\ 0 & 0 & 0 & 16.99 & 0 & 0\\ 0 & 0 & 0 & 0 & 16.99 & 0\\ 0 & 0 & 0 & 0 & 0 & 16.99 \end{array}\right)$$

In the calculation of the VPSC model, the calculation of different loading modes is realized by applying mixed boundary-condition constraints, i.e., the combination of velocity gradient and stress-boundary conditions, as shown in Equation (3). Among them, a is the velocity-gradient loading matrix, b is the velocity gradient control matrix, c is the stress-boundary control matrix, d is the stress boundary matrix, 1 in the matrix means constraint, and 0 means no constraint. The loading condition described by Formula (3) is the boundary condition when unidirectional stretching is along the RD direction. The specific meaning represented is to control the loading velocity-gradient component in the RD (b11) direction to be x, and x is the data obtained in the previous section. In the model calculation, the loaded strain rate is set to 1 s^−1^, as shown in Equation (3)a. No constraints are imposed on other components.
(3)$$\begin{array}{l} a=\left[\begin{array}{ccc} 1 & 1 & 1\\ 1 & 1 & 1\\ 1 & 1 & 1 \end{array}\right],b=\left[\begin{array}{ccc} x & 0 & 0\\ 0 & 0 & 0\\ 0 & 0 & -x \end{array}\right]\\ c=\left[\begin{array}{ccc} 0 & 0 & 0\\ & 0 & 0\\ & & 0 \end{array}\right],d=\left[\begin{array}{ccc} 0 & 0 & 0\\ & 0 & 0\\ & & 0 \end{array}\right] \end{array}$$

In the VPSC model, the hardening law described by Voce hardening is used, and the Neff model for the linear algorithm model is set in the program. The material-hardening parameters of the ZK61 magnesium-alloy rolled plate are determined by fitting the true stress-true strain curve of the ZK61 magnesium-alloy rolled plate stretched along the RD direction and the twin-area fraction. The experimental data are shown above. On one hand, the experimental data can be used to adjust the material-hardening parameters; on the other hand, it can also be used to verify the predicted numerical results. Given that the rolled ZK61 magnesium-alloy plate has a strong basal texture, the hardening parameters of the basal <a> slip and the prismatic <a> slip can be determined for the tensile deformation along the RD direction. Moreover, according to the law of Voce hardening, at a specific temperature, 20 parameters need to be determined for basal <a> slip, prismatic <a> slip, pyramidal <c+a> slip, ETW, and CTW. Considering the complexity of the current deformation under variable temperature and strain-rate conditions, obtaining all material parameters (minimum 60) by fitting stress–strain curves in different directions is impossible. Therefore, the pyramidal <c+a> slip, ETW, and CTW parameters are obtained from the twin-area fraction obtained from the EBSD experimental results as the fitting target. The twin-area fraction here is calculated from the EBSD test results by using Image-Pro Plus 6.0 image-recognition software to count the area fraction of different forms of twins [25].

## 3. Results

### 3.1. Microstructure of ZK61 Magnesium-Alloy Plates of Different Passes

To observe the grain size, texture, and orientation distribution of ZK61 plate after homogenization treatment, EBSD calibration is performed on the sample. A total of 80,000 discrete orientations are measured on the initial plate. Results are shown in Figure 6 for the grain size map, inverse pole Figure (IPF) map, and pole Figure and orientation-difference distribution map of the ZK61 plate after homogenization treatment.

The uniformity of the structure after homogenization heat treatment is improved, and the average grain size is about 48.12 μm. The grain shape and size are relatively uniform, are completely recrystallized, and have a twin-free microstructure. Figure 6 shows that the initial material plate does not show a strong basal texture after processing, the corresponding {0001} pole Figure shows a weakened and diffused basal texture, and the angle of the maximum peak of the misorientation angle is between 0° and 5°. Furthermore, small peaks appear at two typical twin orientation angles, namely, 86.3° {10$\overset{—}{1}$2} tensile twinning and 56.7° {10$\overset{—}{1}$1} compressive twinning. Comparison of the above XRD experimental results reveals that the spectral-line position and relative intensity are consistent with the EBSD test results, and the measured macro-texture and micro-texture are highly consistent. The selected ZK61 magnesium alloy as-cast plate has no obvious preferred orientation after homogenization treatment.

Due to the poor plastic-deformation ability of magnesium alloys at low temperature, symmetric hot rolling is first used for plate rolling, and the rolling temperature is controlled at 380 °C. Heat preservation annealing is needed between passes for 15 min to prevent the occurrence of cracks and prevent subsequent rolling from continuing [26]. If the holding time of annealing is too long, the grains grow up, affecting the analysis of simulation results of the subsequent rolling. The average grain size after the SR-8.6 mm plate of rolling is 7.98 μm, and the average grain size after SR-3.4 mm plate of rolling is 2.34 μm. The evolution process of the microstructure of the rolled plate in each pass is observed. With increased rolling pass, the grain size of the plate gradually decreases.

Figure 7a shows the EBSD-measured IPF maps, (0001) PFs, and misorientation-angle distribution of the SR-8.6 mm plate. According to the statistics of the misorientation-distribution map, the total amount of {10$\overset{—}{1}$2} tensile twins reaches 0.5%, compression twins {10$\overset{—}{1}$1} and {10$\overset{—}{1}$3} reach 1.2% and 1%, respectively, double twins {10$\overset{—}{1}$1}–{10$\overset{—}{1}$2} reach 2.8%, and double twins {10$\overset{—}{1}$3}–{10$\overset{—}{1}$2} reach 3.3%. The microstructure of the plate obviously begins to be refined, and the grain orientation of the plate gradually evolves into a typical basal rolling texture. The new grains in the material are primarily located at the grain boundaries of the elongated grains, and a large number of subgrains exist near the grain boundaries. After experiencing a large rolling strain, the basal texture of the plate rotates greatly, and the maximum pole density is 8.61 mud.

Figure 7b shows the SR-3.4 mm plate of the EBSD-measured IPF maps, (0001) PFs, and misorientation-angle distribution. According to the statistics of the orientation-difference distribution map, the starting amount of tensile twinning {10$\overset{—}{1}$2} is 0.1%, the volume fraction is 0.1%, and the compression twinning {10$\overset{—}{1}$1} and {10$\overset{—}{1}$3} starting amounts are 0.3% and 0.2%, respectively. The double twins {10$\overset{—}{1}$1}–{10$\overset{—}{1}$2} reach 2.1%, and {10$\overset{—}{1}$3}–{10$\overset{—}{1}$2} reaches 3.8%. During rolling, the dominant twin type is double twin, and a typical rolling texture also forms on the basal plane during rolling. The highest strength of the extreme density is 18.10 mud. From the above EBSD test results of the symmetric ly rolled plate, we find that the typical rolling basal texture of the plate appears with the subsequent rolling, and the continuous increase in the extreme density indicates that the basal rolling texture is enhanced.

In the SR-3.4 mm plate, the degree of grain homogeneity and refinement is more obvious. After the subsequent plate is rolled at 280 °C with asymmetric multipass small reduction, we find that the internal microstructure changes. With increased accumulated down pressure during rolling, the number of recrystallization inside the crystal increases, and the grain-size uniformity increases. The average grain sizes are 2.95, 2.87, 2.76, and 2.60 μm for the samples with ASR-1.5 mm, ASR-1.3 mm, ASR-0.75 mm, and ASR-0.5 mm plates, respectively. Considering that the rolling temperature is above the recrystallization temperature of ZK61 magnesium alloy, static recrystallization occurs, causing the grains to grow after rolling. This finding also explains why the grain size of the plate enlarges after multiple passes of asymmetric rolling. However, as the asymmetric rolling progresses, we can still observe that the grain size is still decreasing because recrystallization occurs more fully at high temperature. Consequently, the grain size competes with each other between rolling breakage and grain growth [27,28]. Therefore, as the rolling progresses, the grain size inside the plate becomes increasingly uniform.

Figure 8a shows the EBSD-measured IPF maps, (0001) PFs, and misorientation-angle distribution of the ASR-1.5 mm plate. According to the statistics of the misorientation-distribution map, the volume fraction of the tensile twin {10$\overset{—}{1}$2} initiation amount is 0.1%, the compression twinning {10$\overset{—}{1}$1} and {10$\overset{—}{1}$3} starting amounts are both 0, double twins {10$\overset{—}{1}$1}–{10$\overset{—}{1}$2} is only 0.2%, and {10$\overset{—}{1}$3}–{10$\overset{—}{1}$2} is only 0.6%. The microstructure reveals that the plate structure is obviously refined due to dynamic recrystallization, and the twins basically disappear. Combined with the initial texture shown in the SR-3.4 mm plate of SR, most of the grains rotate from the middle region of the basal plane {0001} pole Figure to the right region during asymmetric rolling. The new grains in the plate are primarily distributed at the grain boundaries of the elongated and inclined grains, and a large number of subgrains exist around the grain boundaries. The deflection angle of the basal texture of the plate relative to the rolling is about 10°, and the maximum strength of the extreme density is 18.26 mud, consistent with the widely reported basal texture formed when the asymmetric rolling temperature is higher than 250 °C. The structure deflection is consistent with.

Figure 8b shows the EBSD-measured IPF maps, (0001) PFs, and misorientation-angle distribution of the ASR-0.75 mm plate. According to the statistics of the misorientation-distribution map, the starting amount of tensile twinning {10$\overset{—}{1}$2} is 0.1%, and the volume fraction is 0.1%, the compression twinning {10$\overset{—}{1}$1} and {10$\overset{—}{1}$3} starting amounts are both 0, the double twins {10$\overset{—}{1}$1}–{10$\overset{—}{1}$2} is only 0.2%, and {10$\overset{—}{1}$3}–{10$\overset{—}{1}$2} is only 0.9%. The microstructure reveals that the plate structure is obviously refined due to dynamic recrystallization, and the twins basically disappear. The maximum strength of its extreme density is 14.94 mud. Compared with the 1.3 mm-thick plate, the texture extreme density decreases, and the texture continues to weaken. Given the asymmetric multipass small reduction rolling, the deflection angle of the texture is reduced compared with the ASR-1.3 mm plate selected previously.

Figure 8c shows the EBSD-measured IPF maps, (0001) PFs, and misorientation-angle distribution of the ASR-0.5 mm plate. The microstructure reveals that the plate structure is obviously refined due to dynamic recrystallization, and the twins basically disappear. The deflection angle of the basal texture of the plate is about 10° relative to the rolling of the ASR-0.75 mm plate. The maximum strength of its extreme density is 16.74 mud. Compared with the increased extreme density of the ASR-0.75 mm plate texture, the basal texture is slightly stronger due to the increase in asymmetric rolling passes.

### 3.2. Analysis of Tensile Test Results and Fracture Morphology Analysis

As shown in Table 3 and Figure 9, the ASR-1.5 mm plate has the best performance during asymmetric rolling. The tensile strength in the RD direction reaches 391.2 MPa, and the tensile strength in the TD direction reaches 398.9 MPa, but the elongation is reduced. After continuous asymmetric rolling, the elongation of the plate increases, but the yield strength and tensile strength decrease. Above room temperature, tensile experiments indicate that the ZK61 magnesium-alloy plate prepared by asymmetric rolling has more uniform grain distribution and finer grain size, deflection of the basal texture, and extreme density than the SR magnesium-alloy plate. The tensile strength and elongation are greatly improved, and the mechanical properties are more excellent.

Table 4 compares the room-temperature properties of ZK60/61 magnesium alloy with different processes. The room-temperature properties YTS and UTS of the plates subjected to this process are better than those of other processes, and the elongation is also better than most of the elongation of ZK60/ZK61 subjected to other processes. The processing technologies with excellent properties in the table are complex and have great limitations, such as equal channel angular extrusion. Considering the production efficiency, processing cost, and product-size constraints, the process used in this study has great potential in practical production.

The key factors that directly affect the properties of plates are grain size and texture. In combination with the changes in grain size and texture in the processing technique discussed above, we can understand the reasons for obtaining high-performance plates through this process.

Figure 10 shows the macro-tensile fracture morphology of the 1.5 mm plate rolled asymmetric ly. The macro-fracture shows that the fracture is uneven, showing tear fracture, and the cavities in the fracture are plastic holes. The three groups of tensile samples have obvious diameter shrinkage, indicating that the fracture of the three groups of samples is ductile fracture. Figure 11 shows the corresponding micro-fracture morphology. A large number of dimples and plastic holes exist on the micro-fracture surface. The dimple size in the TD direction is the most uniform, and the depth is relatively consistent. In the 45° direction, the dimple distribution is uneven, and the size difference is large. In the actual tensile test, the elongation of the sample in the TD direction is also the best, reaching 25.5%, consistent with the observation results.

### 3.3. VPSC Simulation Results

A large number of tensile test data and EBSD test data of different passes and directions are used as reference, so the requirements for parameters in the program are also higher. Nevertheless, fitting multiple experimental data and true stress-strain curves well for a set of material parameters remains very challenging. The Voce hardening model is used to describe the hardening of each deformation mode during the plastic deformation of the material. A total of 20 parameters for the 5 deformation modes involved in the magnesium alloy are determined. Table 5 shows the best hardening parameters determined according to the VPSC simulation results.

From the hardening parameters in Table 3 and Table 5 in the previous section, the results of the stress–strain curve in the RD direction of the VPSC simulation for the 3.4, 1.5, 0.75, and 0.5 mm-thick sampling plates and the experimental results are compared and fitted. Results are shown in Figure 12. Figure 13 compares the VPSC-simulated twinning integral and the twinning area fraction calculated using Image-Pro Plus 6.0 image-recognition software.

## 4. Discussion

This section discusses the starting amount of each slip system for each pass of rolling simulation. The calculation result is obtained from the shear strain generated by each slip system. The total relative starting value of each slip system is 1. Using the material parameters in Table 3 to predict Figure 14, the simulation result of the starting amount of each slip system in the SR-3.4 mm plate rolling simulation is obtained. In the initial stage of deformation, basal <a> slip and compression twins are the main factors, and pyramidal <c+a> slip on the cone surface gradually increases. In the SR-3.4 mm plate rolling deformation, the basal <a> slips in almost all deformation stages. It is the main deformation mode, followed by prismatic <a> slip. The compression twinning {10$\overset{—}{1}$1} system has a low actuation ratio in the simulation. The activation value of {10$\overset{—}{1}$2} tensile twinning is almost zero during the simulated rolling-deformation process.

As shown in Figure 14, using the material parameters in Table 3 to simulate and predict the different asymmetric sampling plates, the change process of the starting amount of the slip system during rolling deformation is determined. Due to the different rolling passes, the accumulated deformation becomes progressively larger, and the slippage and twinning behavior inside the plate obviously differ. Given that the CRSS value of basal <a> slip is much lower than other deformation modes, the microscopic deformation mechanism is dominated by basal <a> slip. Figure 14b,c show that {10$\overset{—}{1}$1} compression twins always start before {10$\overset{—}{1}$2} tensile twins during asymmetric rolling, and it is after {10$\overset{—}{1}$1} compression twins reach a certain activity that {10$\overset{—}{1}$2} tensile twins start to initiate. These findings are consistent with the EBSD experimental observations.

Figure 14c shows that under this pass deformation, {10$\overset{—}{1}$2} tensile twinning and {10$\overset{—}{1}$1} compressive twinning are largely inactive. Figure 14d shows that in this pass rolling, the CRSS value of pyramidal <c+a> slip is higher, the hardening slope is also higher, and pyramidal <c+a> slip basically does not start. During deformation, the pyramidal <c+a> slip and the twinning actuation belong to the competitive mechanism, and the actuation amount of {10$\overset{—}{1}$1} compression twins decreases obviously when the amount of the pyramidal <c+a> slip increases. Whether it is the ASR-1.5 mm or ASR-0.75 mm plate, the pyramidal<c+a> slip starts and participates in the deformation because the shear deformation during asymmetric rolling increases the starting amount of pyramidal <c+a> slip. For the ASR-0.5 mm plate, the amount of downward pressure is too low, and the amount of <c+a> slip start on the cone surface is very small.

## 5. Conclusions

XRD test shows that after the homogenization heat treatment of ZK61 magnesium alloy as-cast plate, the segregation crystallization inside the plate is effectively eliminated. The plate material is uniform, and the strong texture does not appear, laying the foundation for the success of subsequent rolling deformation.

After the subsequent multipass and small-reduction ASR of the six-high ASR mill, we find that the mechanical properties of the sampled ASR-1.5 mm plate are significantly improved. The tensile strength in the RD direction reaches 391.2 MPa, and the tensile strength in the TD direction reaches 398.9 MPa, but the elongation is reduced. After continuous asymmetric rolling, plate elongation increases, but the yield strength and tensile strength decrease. The basal texture is obviously shifted to the rolling direction, the offset angle is ~10°, the strength of the basal texture is continuously weakened, and the grains are continuously refined.

The microstructure hardening parameters of the plates with good fitting are obtained through the multiscale simulation, in which the obtained simulated stress–strain curve is fitted with the stress–strain curve obtained from the room-temperature tensile experiment. The twinning data obtained by EBSD detection are compared with the simulated twinning data. The VPSC simulation results show that during the SR, the fourth and fifth passes are at the initial stage of rolling deformation, the basal <a> slip is the main deformation system, and the prismatic <a> slip and the pyramidal <c+a> slip are the supplement. As the amount of deformation increases, pyramidal <c+a> slip and basal <a> slip form a competitive relationship. The slip initiation amount of the cone pyramidal <c+a> is greater than that of the basal <a> slip, and the {10$\overset{—}{1}$1} compression twinning initiation amount is limited. The starting amounts of the slip system in the sixth pass and SR-3.4 mm plate are similar. All of them are primarily based on basal <a> slip, supplemented by prismatic <a>. Pyramidal <c+a> has limited slip start amount. The activation amount of {10$\overset{—}{1}$1} compression twinning is large at the initial stage of deformation, and the activation ratio gradually decreases with increased deformation. In asymmetric rolling, the basal <a> slip is dominant, pyramidal <c+a> slip contribution comes second, and the amount of deformation involved in the twin system is relatively limited.

## Figures and Tables

**Figure 1 materials-16-01320-f001:**
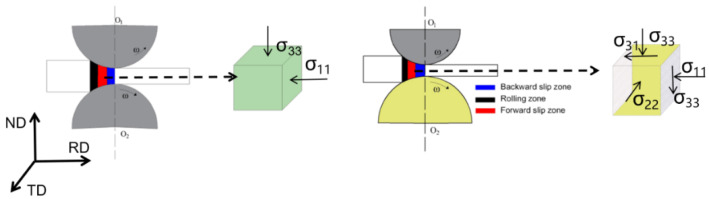
Schematic diagram of the stress state in the rolling experiment.

**Figure 2 materials-16-01320-f002:**
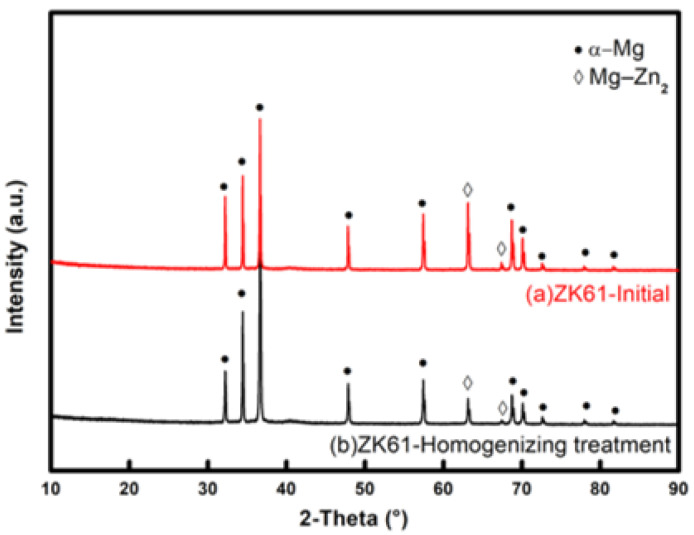
X-ray diffraction pattern of ZK61 magnesium alloy before (**a**) and after (**b**) homogenization treatment.

**Figure 3 materials-16-01320-f003:**
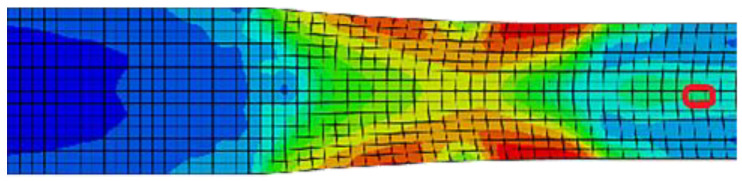
Selected element(middle part of red circle).

**Figure 4 materials-16-01320-f004:**
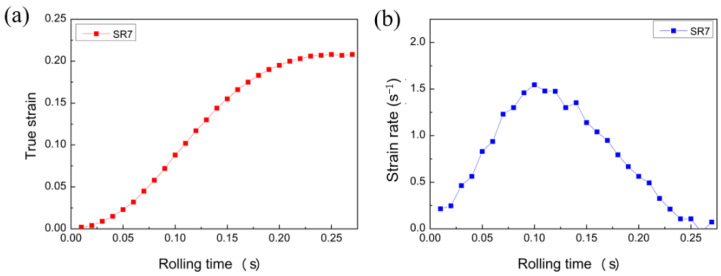
(**a**) change in stress field, (**b**) evolution of strain rate.

**Figure 5 materials-16-01320-f005:**
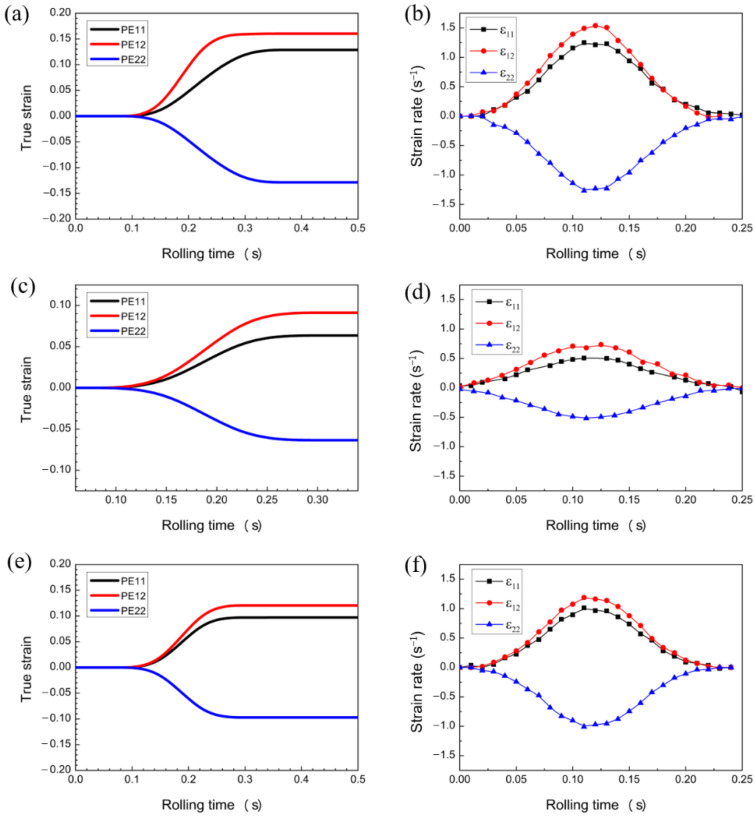
Evolution of equivalent stress field and strain rate in the center of sample plate by asymmetric rolling: changes in the (**a**) 1.5 mm stress field, (**b**) 1.5 mm strain rate, (**c**) 0.75 mm stress field, (**d**) 0.75 mm strain rate, (**e**) 0.5 mm stress field, and (**f**) 0.5 mm strain rate.

**Figure 6 materials-16-01320-f006:**
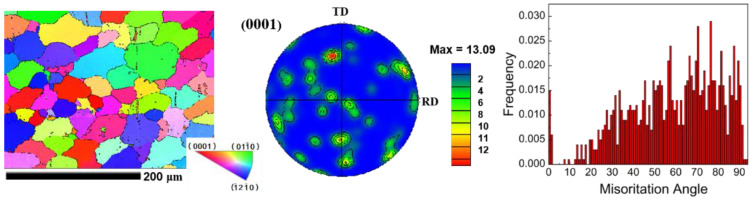
Initial plate EBSD-measured IPF maps, (0001) PFs, and misorientation-angle distribution.

**Figure 7 materials-16-01320-f007:**
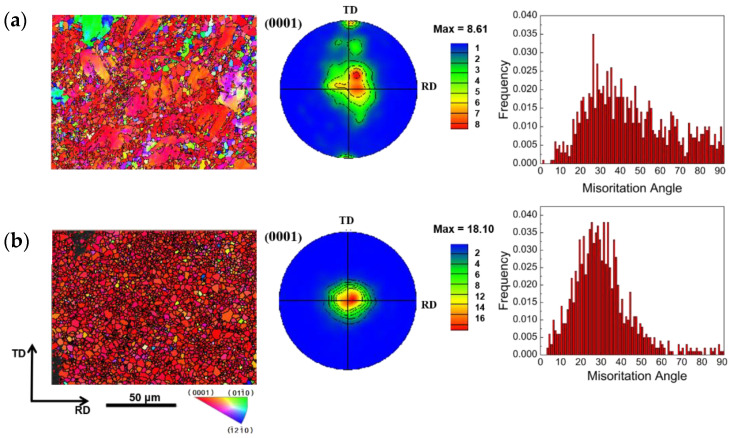
EBSD IPF maps, (0001) PFs, and misorientation-angle distribution of SR: (**a**) SR-8.6 mm, (**b**) SR-3.4 mm.

**Figure 8 materials-16-01320-f008:**
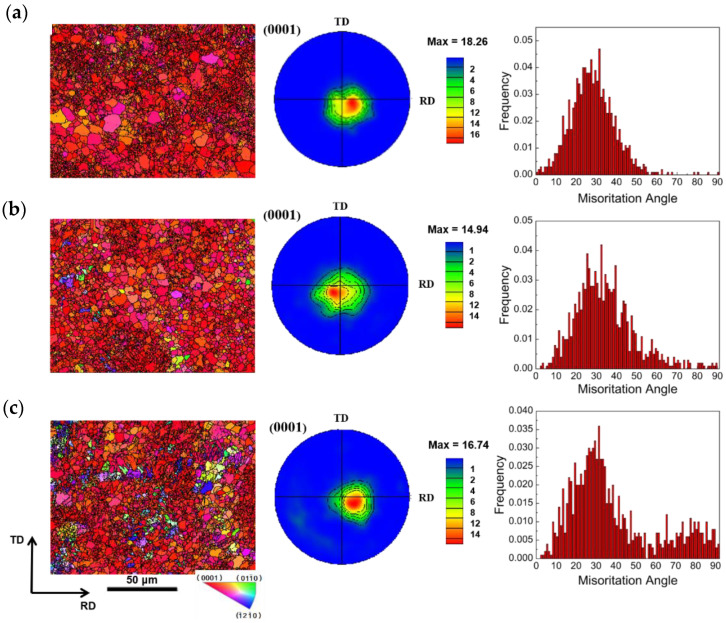
EBSD-measured IPF maps, (0001) PFs, and misorientation-angle distribution of ASR: (**a**) ASR-1.5 mm, (**b**) ASR-0.75 mm, (**c**) ASR-0.5 mm.

**Figure 9 materials-16-01320-f009:**
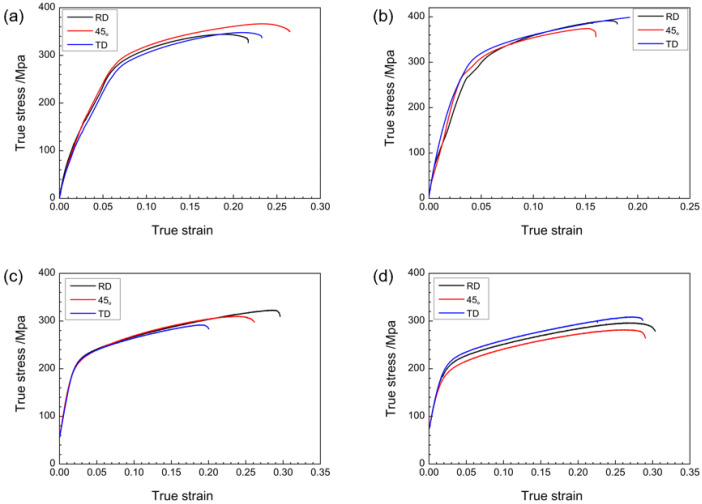
Tensile curves at room temperature along RD, 45°, and TD directions of sampled plates: (**a**) SR-3.4 mm plate, (**b**) ASR-1.5 mm plate, (**c**) ASR-0.75 mm plate, and (**d**) ASR-0.5 mm plate.

**Figure 10 materials-16-01320-f010:**
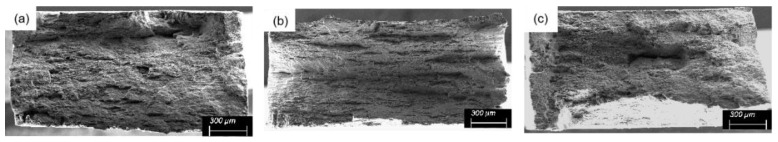
Macro-tensile fracture morphology of asymmetric rolled 1.5 mm plate: (**a**) 45° direction, (**b**) RD direction, and (**c**) TD direction.

**Figure 11 materials-16-01320-f011:**
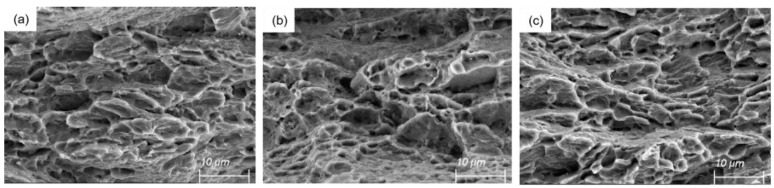
Micro-tensile fracture morphology of asymmetric rolled 1.5 mm plate: (**a**) 45° direction, (**b**) RD direction, and (**c**) TD direction.

**Figure 12 materials-16-01320-f012:**
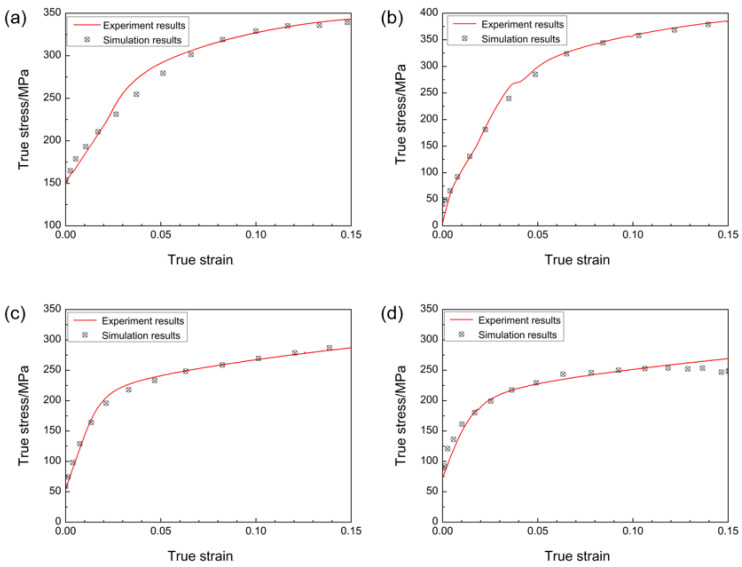
Experimental and simulation comparison of stress–strain curves of sampling plate by asymmetric rolling: (**a**) SR-3.4 mm plate, (**b**) ASR-1.5 mm plate, (**c**) ASR-0.75 mm plate, and (**d**) ASR-0.5 mm plate.

**Figure 13 materials-16-01320-f013:**
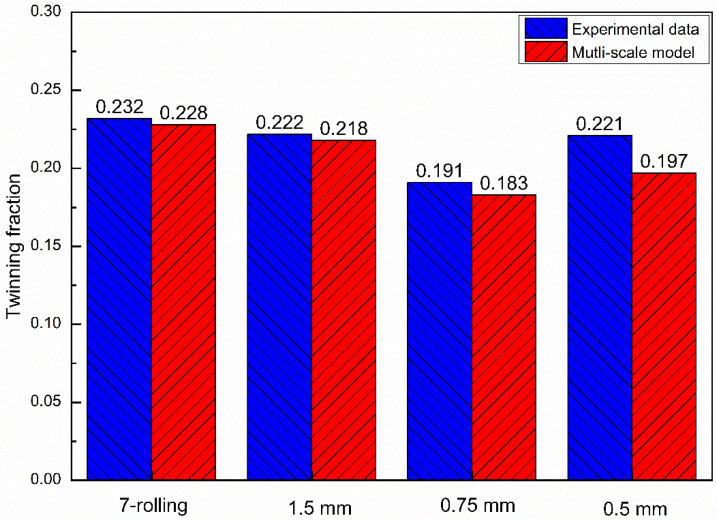
Comparison of twin-plane integral number of VPSC-simulated asymmetric rolling plate with that obtained from EBSD statistics.

**Figure 14 materials-16-01320-f014:**
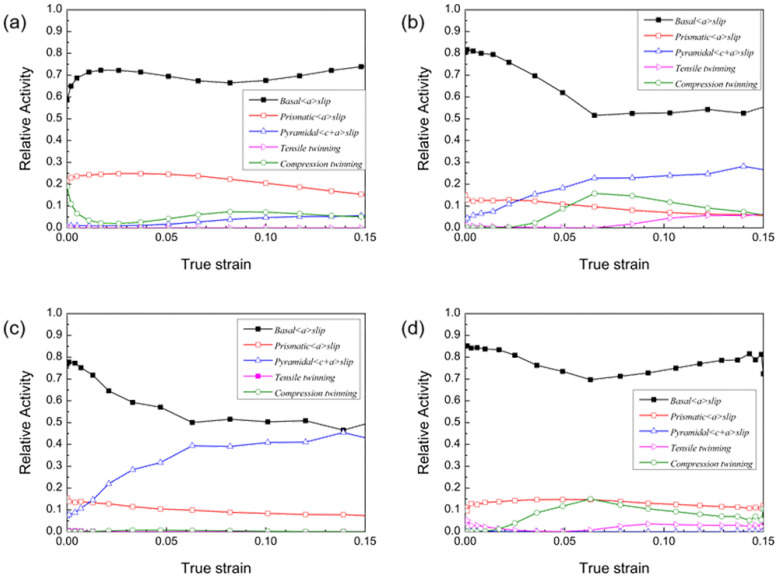
Selection of starting quantity of each slip system for asymmetric rolling: (**a**) SR-3.4 mm plate, (**b**) ASR-1.5 mm plate, (**c**) ASR-0.75 mm plate, and (**d**) ASR-0.5 mm plate.

**Table 1 materials-16-01320-t001:** Chemical composition of ZK61 magnesium alloy.

Alloying element	Mn	Zr	Si	Zn	Mg
Element content (wt.%)	<0.02	0.76	<0.02	6.6	Balance

**Table 2 materials-16-01320-t002:** Multipass-rolling experiment of ZK61 magnesium-alloy billet.

Run Pass	Thickness of Plate before Rolling (mm)	Thickness of Plate after Rolling (mm)	Single Pressure Down (%)
1	17.0	15.0	12.0
2	15.0	11.5	23.3
3	11.5	8.6	25.2
4	8.6	6.9	19.6
5	6.9	5.6	18.8
6	5.6	4.5	19.8
7	4.5	3.4	24.4
8–20	3.4	0.5	10.0

**Table 3 materials-16-01320-t003:** Mechanical properties of plates sampled in different passes.

Tensile Samples		$\sigma_{s}$/MPa	$\sigma_{b}$/MPa	δ/%	$\sigma_{s}$ $/\sigma_{b}$
SR-3.4 mm plate	RD	230.2	344.3	21.7	0.67
	45°	253.7	366.3	26.5	0.69
	TD	233.3	348.0	23.3	0.67
ASR-1.5 mm plate	RD	264.4	391.2	19.8	0.68
	45°	273.2	374.3	17.4	0.73
	TD	290.4	398.9	25.5	0.73
ASR-0.75 mm plate	RD	198.4	322.6	34.7	0.62
6.	45°	198.4	310.1	30.1	0.64
	TD	198.4	291.9	22.4	0.68
ASR-0.5 mm plate	RD	183.1	296.4	35.6	0.62
	45°	165.3	281.5	33.8	0.59
	TD	196.2	308.6	33.6	0.64

**Table 4 materials-16-01320-t004:** Comparison of room-temperature properties of ZK60/ZK61 alloy under different processes.

Processing Method	YTS (Mpa)		UTE (Mpa)		Elongation (%)		References
	RD	TD	RD	TD	RD	TD	
Cold rolling + annealing at 250 °C	241.8	254.9	285.1	293.6	26.9	27.3	[12]
Hot rolling at 400 °C	215.0	196.4	304.6	296.3	20.8	21.7	[29]
Welding + cold rolling	168	191	297	301	23.6	24.4	[30]
Rolling at 250 °C	292.8	315.2	361.4	367.7	8.1	18.8	[31]
Hot extrusion at 380 °C + rolling at 250 °C	210	193	309	298	20.3	25.9	[32]
Rolling at 250 °C	280.1	218.2	358.9	328.7	16.2	16.7	[33]
asymmetric reduction rolling	200		264		27		[34]
Equal channel extrusion + cold rolling			396		9.4		[35]
Cold rolling	242	256	284	295	28	26	[36]
Rolling state	226	178	315	323	14	20	[37]
symmetric hot rolling + asymmetric warm rolling	264.4	290.4	391.2	398.9	19.8	25.5	Present work

**Table 5 materials-16-01320-t005:** VPSC simulation for the best fitting hardening parameters in asymmetric rolling.

Plate	Deformation Mode	$\tau_{0}$	$\tau_{1}$	$\theta_{0}$	$\theta_{1}$
SR-3.4 mm plate	Basal<a>	21	5	40	5
	Prismatic<a>	48	25	110	10
	Pyramidal<c+a>	115	40	800	70
	Tensile twinning	140	30	180	60
	Compression twinning	90	40	1500	100
ASR-1.5 mm plate	Basal<a>	6	20	230	80
	Prismatic<a>	12	60	710	90
	Pyramidal<c+a>	30	120	1800	75
	Tensile twinning	8	20	240	150
	Compression twinning	70	40	2500	250
ASR-0.75 mm plate	Basal<a>	8	10	300	80
	Prismatic<a>	20	32	1200	100
	Pyramidal<c+a>	40	60	2500	50
	Tensile twinning	10	15	300	150
	Compression twinning	80	21	3200	300
ASR-0.5 mm plate	Basal<a>	20	10	400	3
	Prismatic<a>	45	32	1200	8
	Pyramidal<c+a>	100	100	2500	10
	Tensile twinning	25	0	0	120
	Compression twinning	125	15	30	145

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
