# Peer review of "Plastic Deformation Mechanism of High Strength and Toughness ZK61 Magnesium Alloy Plate by Multipass Horizontal Continuous Rolling"

_materials, 2023, doi:10.3390/ma16031320_

Round 1
Reviewer 1 Report
You have obtained new and important results presentation is clear..
+. Yo should describe the equipment LDK (line (94).
2. You could describe abbreviation SR (l. 99).
3. Can you determined 00.1 +-1 microm?
4.. Accuracy of stress?
More detailed comments.
1) The effect of a symmetric and an antisymitic rolling on the microstructure and mechanical properties of Mg alloys ZK61 has been investigated.
2) The topics is important - activity of twins after different rolling passes (grain refinment etc).
5) Conclusions are based on the results.
6) The references are appropriate.
Reviewer 2 Report
In a peer-reviewed paper, large-scale studies of the effect of asymmetric rolling on the formation of the microstructure and properties of rolled products from magnesium alloy ZK61 were carried out. However, it is a matter of great regret that the authors did not accurately present the obtained results in paper. Now the paper is overloaded with information (although the size of the article is not very large, only 17 pages) and it is difficult for the reader to understand. I will point out some points that the authors should pay attention to and make changes.
1. Section 2 Experimental requires clarification of the methodology of experimental studies. I recommend to bring the graph (scheme) of experimental rolling. So that the reader can quickly understand to what sizes, at what temperatures and with what degrees of deformation the sheet was rolled.
2. Line 91 does not specify the dimensions of the cast plates.
3. Line 99 says "plate is first rolled by SR". It is necessary to decipher what kind of method it is.
4. At what temperature were samples rolled 1.5; 1.3; 0.75 mm. What is the temperature of the samples? What are these samples? Further studies indicate 3.4; 1.5; 0.75 and 0.5 mm.
5. Table 2 shows the individual degrees of deformation and sheet thickness. In this case, you must specify the same number of characters after the ".". It is necessary to correct the values "15.0", "17.0"; "12.0" and "10.0".
6. When studying the rolling process, and especially cold plastic deformation, it is very important to talk about the total degree of deformation. In this work, the total deformation is not discussed. You need to add this information. What are the values of the total deformation of samples 3.4; 1.5; 0.75 and 0.5 mm?
7. In Fig. 9 and 12 on the x-axis is the amount of deformation in % or in fractions of "1"?
8. In Figure 5, the caption says "0.75 mm strain rate". The strain rate is indicated in s-1. What is said here? The captions may have been worded incorrectly.
9. Table 4 (excluding your results), in my opinion, is better presented in the "Introduction".
10. What is the rolling speed for experiment and simulation? This is also important in the study of properties and microstructure.
Reviewer 3 Report
This paper focused on Plastic deformation mechanism of high strength and toughness ZK61 magnesium alloy plate by multipass horizontal continuous rolling. The following points need to be clarified for paper to be considered for the publication:
- It is suggested that the authors should modify the last part of the introduction to: 1) clearly mention the goal and novelty of this work, 2) mention the methodology used and its important to place the major hypothesis, 3) mention the structure of the paper or procedure of their work and expected results.
- Details about the two-dimensional FEM should be given. Element types used, BCs, analysis type, which software etc.
- Fig. 3 is not self-explanatory. Which values do the colours represent?
- The figures have no standard deviations? How many times each test was repeated? That needs to be clarified.
- Some very recent studies could be added in the literature review: https://doi.org/10.1007/s11661-022-06846-4, https://doi.org/10.3390/cryst13010137,
- The paper lacks a discussion on the weakness and limitations of the present methodology/study.
- In the discussion part of the paper (Section 4), the findings should be analyzed/discussed with the studies from the literature review.
- The conclusion of the paper is too long. It needs to be shortened, only main conclusion drawn should be placed.
Typos
Line 175 “1 s–1” should read “1 s–1”
Line 343 “VPSC aimulation results” should read “VPSC Simulation results”
Round 2
Reviewer 2 Report
The authors made responses to my comments and corrected the paper. I recommend publishing the article in this version.
Reviewer 3 Report
The paper is revised as requested.